# $\beta$-risk: a New Surrogate Risk for Learning from Weakly Labeled Data

**Valentina Zantedeschi**[*]  **Rémi Emonet**

**Marc Sebban**

firstname.lastname@univ-st-etienne.fr
Univ Lyon, UJM-Saint-Etienne, CNRS, Institut d Optique Graduate School,
Laboratoire Hubert Curien UMR 5516, F-42023, SAINT-ETIENNE, France

## Abstract

During the past few years, the machine learning community has paid attention to developing new methods for learning from weakly labeled data. This field covers different settings like semi-supervised learning, learning with label proportions, multi-instance learning, noise-tolerant learning, etc. This paper presents a generic framework to deal with these weakly labeled scenarios. We introduce the $\beta$-risk as a generalized formulation of the standard empirical risk based on surrogate margin-based loss functions. This risk allows us to express the reliability on the labels and to derive different kinds of learning algorithms. We specifically focus on SVMs and propose a soft margin $\beta$-SVM algorithm which behaves better that the state of the art.

## 1   Introduction

The growing amount of data available nowadays allowed us to increase the confidence in the models induced by machine learning methods. On the other hand, it also caused several issues, especially in supervised classification, regarding the availability of labels and their reliability. Because it may be expensive and tricky to assign a reliable and unique label to each training instance, the data at our disposal for the application at hand are often *weakly labeled*. Learning from weak supervision has received important attention over the past few years [14, 12]. This research field includes different settings: only a fraction of the labels are known (Semi-Supervised learning [22]); we can access only the proportions of the classes (Learning with Label Proportions [19] and Multi-Instance Learning [8]); the labels are uncertain or noisy (Noise-Tolerant Learning [1, 18, 16]); different discording labels are given to the same instance by different experts (Multi-Expert Learning [21]); labels are completely unknown (Unsupervised Learning [11]). As a consequence of this statement of fact, the data provided in all these situations cannot be fully exploited using supervised techniques, at the risk of drastically reducing the performance of the learned models. To address this issue, numerous machine learning methods have been developed to deal with each of the previous specific situations. However, all these weakly labeled learning tasks share common features mainly relying on the confidence in the labels, opening the door to the development of generic frameworks. Unfortunately, only a few attempts have tried to address several settings with the same approach. The most interesting one has been presented in [14] where the authors propose WELLSVM which is dedicated to deal with three different weakly labeled learning scenarios: semi-supervised learning, multi-instance learning and clustering. However, WELLSVM focuses specifically on Support Vector Machines and it requires to

---

[*]http://vzantedeschi.com/

derive a new optimization problem for each new task. Even though WELLSVM constitutes a step further towards general models, it stopped in midstream constraining the learner to use SVMs.

This paper aims to bridge this gap by presenting a generic framework for learning from weakly labeled data. Our approach is based on the derivation of the $\beta$-risk , a new surrogate empirical risk defined as a strict generalization of the standard empirical risk relying on surrogate margin-based loss functions. The main interesting property of the $\beta$-risk comes from its ability to exploit the information given by the weakly supervised setting and encoded as a $\beta$ matrix reflecting the supervision on the labels. Moreover, the instance-specific weights $\beta$ let one integrate in classical methods the side information provided by the setting. This is the peculiarity w.r.t. [18, 16]: in both papers, the proposed losses are defined using class-dependent weights (fixed to 1/2 for the first paper, and dependent on the class noise rate for the latter) while in our approach the used weights are provided for each instance, which gives a more flexible formulation. Making use of this $\beta$-risk , we design a generic algorithm devoted to address different kinds of aforementioned weakly labeled settings. To allow a comparison with the state of the art, we instantiate it with a learner that takes the form of an SVM algorithm. In this context, we derive a soft margin $\beta$-SVM algorithm and show that it outperforms WELLSVM.

The remainder of this paper is organized as follows: in Section 2, we define the empirical surrogate $\beta$-risk and show under which conditions it can be used to learn without explicitly accessing the labels; we also show how to instantiate $\beta$ according to the weakly labeled learning setting at hand; in Section 3, we present our generic iterative algorithm for learning with weakly labeled data and in Section 4 we exploit our new framework to derive a novel formulation of the Support Vector Machine problem, the $\beta$-SVM ; finally, we report experiments in semi-supervised learning and learning with label noise, conducted on classical datasets from the UCI repository [15], in order to compare our algorithm with the state of the art approaches.

## 2 From Classical Surrogate Losses and Surrogate Risks to the $\beta$-risk

In this section, we first provide reminders about surrogate losses and then exploit the characteristics of the popular loss functions to introduce the empirical surrogate $\beta$-risk . The $\beta$-risk formulation allows us to tackle the problem of learning with weakly labeled data. We show under which conditions it can be used instead of the standard empirical surrogate risk (defined in a fully supervised context). Those conditions give insight on how to design algorithms that learn from weak supervision. We restrain our study to the context of binary classification.

### 2.1 Preliminaries

In statistical learning, a common approach for choosing the optimal hypothesis $h^*$ from a hypothesis class $\mathcal{H}$ is to select the classifier that minimizes the expected risk over the joint space $Z = X \times Y$, where $X$ is the feature space and $Y$ the label space, expressed as

$$\mathcal{R}_\ell(h) = \int_{X \times Y} \ell(yh(x))p(x,y)dxdy$$

with $\ell : \mathcal{H} \times Z \to \mathbb{R}^+$ a margin-based loss function.

Since the true distribution of the data $p(x,y)$ is usually unknown, machine learning algorithms typically minimize the empirical version of the risk, computed over a finite set $\mathcal{S}$ composed of $m$ instances $(x_i, y_i)$ i.i.d. drawn from a distribution over $X \times \{-1,1\}$:

$$\mathcal{R}_\ell(\mathcal{S}, h) = \frac{1}{m} \sum_{i=1}^{m} \ell(y_i h(x_i)).$$

The most natural loss function is the so-called 0-1 loss. As this function is not convex, not differentiable and has zero gradient, other loss functions are commonly employed instead. These losses, such as the logistic loss (e.g., for the logistic regression [6]), the exponential loss (e.g., for boosting techniques [10]) and the hinge loss (e.g., for the SVM [7]), are convex and smooth relaxations of the 0-1 loss. Theoretical studies on the characteristics and behavior of such *surrogate losses* can be found in [17, 2, 20]. In particular, [17] showed that each commonly used surrogate loss can be

characterized by a permissible function $\phi$ (see below) and rewritten as $F_\phi(x)$

$$F_\phi(x) = \frac{\phi^*(-x) - a_\phi}{b_\phi}$$

where $\phi^*(x) = \sup_a(xa - \phi(a))$ is the Legendre conjugate of $\phi$ (for more details, see [4]), $a_\phi = -\phi(0) = -\phi(1) \geq 0$ and $b_\phi = -\phi(\frac{1}{2}) - a_\phi > 0$. As presented by the authors of [13] and [17], a permissible function is a function $f : [0,1] \to \mathbb{R}^-$, symmetric about $-\frac{1}{2}$, differentiable on $]0,1[$ and strictly convex. For instance, the permissible function $\phi_{log}$ related to the logistic loss $F_\phi(x) = \log(1 + \exp^{-x})$ is:

$$\phi_{log}(x) = x \log(x) + (1 - x) \log(1 - x)$$

and $a_\phi = 0$ and $b_\phi = \log(2)$.

As detailed in [17], considering a surrogate loss $F_\phi$, the empirical surrogate risk of an hypothesis $h : X \to \mathbb{R}$ w.r.t. $\mathcal{S}$ can be expressed as:

$$\mathcal{R}_\phi(\mathcal{S}, h) = \frac{1}{m} \sum_{i=1}^m D_\phi\big(y_i, \nabla_\phi^{-1}(h(x_i))\big) = \frac{b_\phi}{m} \sum_{i=1}^m F_\phi(y_i h(x_i))$$

with $D_\phi$ the Bregman Divergence

$$D_\phi(x, y) = \phi(x) - \phi(y) - (x - y)\nabla_\phi(y).$$

In order to evaluate such risk $\mathcal{R}_\phi(\mathcal{S}, h)$, it is mandatory to provide the labels $y$ for all the instances. In addition, it is not possible to take into account eventual uncertainties on the given labels. Consequently, $\mathcal{R}_\phi$ is defined in a totally supervised context, where the labels $y$ are known and considered to be true. In order to face the numerous situations where training data may be weakly labeled, we claim that there is a need to fill the gap by defining a new empirical surrogate risk that can deal with such settings. In the following section, we propose a generalization of the empirical surrogate risk, called the empirical surrogate $\beta-$risk, which can be employed in the context of weakly labeled data instead of the standard one under some linear conditions on the margin.

## 2.2 The Empirical Surrogate $\beta$-risk

Before defining the empirical surrogate $\beta$-risk for any loss $F_\phi$ and hypothesis $h \in \mathcal{H}$, let us rewrite the definition of $\mathcal{R}_\phi$ introducing a new set of variables named $\beta$, and that can be laid out as a $2 \times m$ matrix.

**Lemma 2.1.** *For any $\mathcal{S}$, $\phi$ and $h$, and for any non-negative real coefficients $\beta_i^{-1}$ and $\beta_i^{+1}$ defined for each instance $x_i \in \mathcal{S}$ such that $\beta_i^{-1} + \beta_i^{+1} = 1$, the empirical surrogate risk $\mathcal{R}_\phi(\mathcal{S}, h)$ can be rewritten as*

$$\mathcal{R}_\phi(\mathcal{S}, h) = \mathcal{R}_\phi(\mathcal{S}, h, \beta)$$

*where*

$$\mathcal{R}_\phi(\mathcal{S}, h, \beta) = \frac{b_\phi}{m} \sum_{i=1}^m \sum_{\substack{\sigma \in \\ \{-1,+1\}}} \beta_i^\sigma F_\phi(\sigma h(x_i)) + \frac{1}{m} \sum_{i=1}^m \beta_i^{-y_i}(-y_i h(x_i)).$$

The coefficient $\beta_i^{+1}$ (resp. $\beta_i^{-1}$) for an instance $x_i$ can be interpreted here as the degree of confidence in (or the probability of) the label +1 (resp. -1) assigned to $x_i$.

*Proof.*

$$\mathcal{R}_\phi(\mathcal{S}, h) = \frac{b_\phi}{m} \sum_{i=1}^{m} F_\phi(y_i h(x_i))$$

$$= \frac{b_\phi}{m} \sum_{i=1}^{m} \left( \beta_i^{y_i} F_\phi(y_i h(x_i)) + \beta_i^{-y_i} F_\phi(y_i h(x_i)) \right) \tag{1}$$

$$= \frac{b_\phi}{m} \sum_{i=1}^{m} \left( \beta_i^{y_i} F_\phi(y_i h(x_i)) + \beta_i^{-y_i} \left( F_\phi(-y_i h(x_i)) - \frac{y_i h(x_i)}{b_\phi} \right) \right) \tag{2}$$

$$= \frac{b_\phi}{m} \sum_{i=1}^{m} \sum_{\substack{\sigma \in \\ \{-1,+1\}}} \beta_i^\sigma F_\phi(\sigma h(x_i)) + \frac{1}{m} \sum_{i=1}^{m} \beta_i^{-y_i}(-y_i h(x_i)). \tag{3}$$

Eq. (1) is because $\beta_i^{-1} + \beta_i^{+1} = 1$; Eq. (2) is due to the fact that $\phi^*(-x) = \phi^*(x) - x$ (see the supplementary material) for any permissible function $\phi$, so that $F_\phi(x) = \frac{\phi^*(-x) - a_\phi}{b_\phi} = \frac{\phi^*(x) - a_\phi - x}{b_\phi} = F_\phi(-x) - \frac{x}{b_\phi}$ . $\qquad\square$

From Eq. (3), and considering that the sample $\mathcal{S}$ is composed by the finite set of features $\mathcal{X}$ and labels $\mathcal{Y}$, we can write that

$$\mathcal{R}_\phi(\mathcal{S}, h) = \mathcal{R}_\phi(\mathcal{S}, h, \beta) = \mathcal{R}_\phi^\beta(\mathcal{X}, h) - \frac{1}{m} \sum_{i=1}^{m} \beta_i^{-y_i} y_i h(x_i) \tag{4}$$

where

$$\mathcal{R}_\phi^\beta(\mathcal{X}, h) = \frac{b_\phi}{m} \sum_{i=1}^{m} \sum_{\substack{\sigma \in \\ \{-1,+1\}}} \beta_i^\sigma F_\phi(\sigma h(x_i))$$

is the empirical surrogate $\beta$-risk for a matrix $\beta = [\beta_0^{+1}, ..., \beta_m^{+1} | \beta_0^{-1}, ..., \beta_m^{-1}]$.

It is worth noticing that $\mathcal{R}_\phi(\mathcal{S}, h, \beta)$ is expressed in the form of a sum of two terms: the second one takes into account the labels of the data, while the first one, the $\beta$-risk, focuses on the loss suffered by $h$ over $\mathcal{X}$ without explicitly needing the labels $\mathcal{Y}$.

The empirical $\beta$-risk is a generalization of the empirical risk: setting $\beta_i^{y_i} = 1$ (and thus $\beta_i^{-y_i} = 0$) for each instance, the second term vanishes and we retrieve the classical formulation of the empirical risk. Additionally, as developed in Section 2.3, the introduction of $\beta$ makes it possible to inject some side-information about the labels. For this reason, we claim that the $\beta$-risk is suited to deal with classification in the context of weakly labeled data.

Let us now focus on the conditions allowing the empirical $\beta$-risk (i) to be a surrogate of the 0-1 loss-based empirical risk and (ii) to be sufficient to learn with a weak supervision on the labels. From (4), we deduce:

$$\mathcal{R}_\phi^\beta(\mathcal{X}, h) = \mathcal{R}_\phi(\mathcal{S}, h, \beta) + \frac{1}{m} \sum_{i=1}^{m} \beta_i^{-y_i} y_i h(x_i) \geq \mathcal{R}_{0/1}(\mathcal{S}, h) + \frac{1}{m} \sum_{i=1}^{m} \beta_i^{-y_i} y_i h(x_i) \tag{5}$$

where $\mathcal{R}_{0/1}(\mathcal{S}, h)$ the empirical risk related to the 0-1 loss and Eq. (5) is because $b_\phi F_\phi(x) \geq F_{0/1}(x)$ (for any surrogate loss).

It is possible to ensure that the $\beta$-risk is both a convex upper-bound of the 0-1 loss based risk and a relaxation as tight as the traditional risk (i.e., that we have $\mathcal{R}_{0/1}(\mathcal{S}, h) \leq \mathcal{R}_\phi^\beta(\mathcal{X}, h) \leq \mathcal{R}_\phi(\mathcal{S}, h))$ is to force the following constraint: $\sum_{i=1}^{m} \beta_i^{-y_i} y_i h(x_i) = 0$.

Unfortunately, the constraint $\sum_{i=1}^{m} \beta_i^{-y_i} y_i h(x_i) = 0$ still depends on the vector $y$ of labels, which is not always provided and most likely uncertain or inaccurate in a weakly labeled data setting. We will show in Section 3 that this issue can be overcome by means of an iterative 2-step learning procedure, that first learns a classifier minimizing the $\beta$-risk , possibly violating the constraint, and then learns a new matrix $\beta$ that fulfills the constraint.

## 2.3 Instantiating $\beta$ for Different Weakly Supervised Settings

The $\beta$-risk can be used as the basis for handling different learning settings, including weakly labeled learning. This can be achieved by fixing the $\beta$ values, choosing their initial values or putting a prior on them. We have already seen that, fully supervised learning can be obtained by fixing all $\beta$ values to 1 for the assigned class and to 0 for the opposite class. The current section provides guidance on how $\beta$ could be instantiated to handle various weakly labeled settings.

In a *semi-supervised* setting, as detailed in the experimental section, we propose to initialize the $\beta$ of unlabeled points to $0.5$ and then to automatically refine them in an iterative process. Going further, and if we are ready to integrate spatial or topological information in the process, the $\beta$ values of each unlabeled point could be initialized using a density estimation procedure (e.g., by considering the label proportions of the $k$ nearest labeled neighbors). In the context of *multi-expert learning*, the experts' votes for each instance $i$ can simply be averaged to produce the $\beta_i$ values (or their initialization, or a prior). The case of *learning with label proportions* is especially useful for privacy-preserving data processing: the training points are grouped into bags and, for each bag, the proportion of labels are given. One way of handling such supervision is to initialize, for each bag, all the $\beta$ with the same value that corresponds to the provided proportion of labels. *Noise-tolerant learning* aims at learning in the presence of label noise, where labels are given but can be wrong. For any point that can be possibly noisy, a direct approach is to use lower $\beta$ values (instead of 1 in the supervised case) and refine them as in the semi-supervised setting. $\beta$ can also be initialized using the label proportion of the $k$ nearest labeled example (as done in the experimental section). The case of *Multiple Instance Learning* (MIL) is trickier: in a typical MIL setting, instances are grouped in bags and the supervision is given as a single label per bag that is positive if the bag contains at least one positive instance (negative bags contain only negative instances). A straightforward solution would be to recast the MIL supervision as a "learning with label proportion" (e.g., considering exactly one positive instance in each bag). It is not fully satisfying and a more promising solution would be to consider, within each bag, the set of $\beta^{+1}$ variables and put a sparsity-inducing prior on them. This approach would be a less-constrained version of the relaxation proposed in WellSVM [14] (where it is supposed that there is exactly one positive instance per positive bag) and could be achieved by a $l_1$ penalty or using a Dirichlet prior (with low $\alpha$ to promote sparsity).

## 3 An Iterative Algorithm for Weakly-labeled Learning

As explained in Section 2, a sufficient condition for guaranteeing that the $\beta$-risk is a convex upper-bound of the 0-1 loss based risk and it is not worse than the traditional risk is to fix $\sum_{i=1}^{m} \beta_i^{-y_i} y_i h(x_i) = 0$. However, the previous constraint depends on the labels. We overcome this problem by (i) iteratively learning a classifier minimizing the $\beta$-risk and most likely violating the constraint and then (ii) learning a new matrix $\beta$ that fulfills it. The algorithm is generic. It can be used in different weakly labeled settings and can be instantiated with different losses and regularizations, as we will do in the next Section with SVMs.

As the process is iterative, let $^t\beta$ be the estimation of $\beta$ at iteration $t$. At each iteration, our algorithm consists in two steps. We first learn an hypothesis $h$ for the following problem $P_1$:

$$h^{t+1} = P_1(\mathcal{X}, {}^t\beta) = \arg\min_h c\mathcal{R}_\phi^{{}^t\beta}(\mathcal{X}, h) + \mathcal{N}(h)$$

which boils down to minimizing the $\mathcal{N}$-regularized empirical surrogate $\beta$-risk over the training sample $\mathcal{X}$ of size $m$, where $\mathcal{N}$, for instance, can take the form of a $L_1$ or a $L_2$ norm.

Then, we find the optimal $\beta$ of the following problem $P_2$ for the points of $\mathcal{X}$:

$$^{t+1}\beta = P_2(\mathcal{X}, h^{t+1}) = \arg\min_\beta \mathcal{R}_\phi^\beta(\mathcal{X}, h^{t+1})$$

$$s.t. \sum_{i=1}^{m} \beta_i^{-y_i}(-y_i\, h^{t+1}(x_i)) = 0$$

$$\beta_i^{-1} + \beta_i^{+1} = 1, \ \ \beta_i^{-1} \geq 0, \beta_i^{+1} \geq 0 \ \ \forall i = 1..m\,.$$

For this step, a vector of labels is required. We choose to re-estimate it at each iteration according to the current value of $\beta$: we affect to an instance the most probable label, i.e. the $\sigma$ corresponding

to the biggest $\beta^\sigma$. The matrix $\beta$ has to be initialized at the beginning of the algorithm according to the problem setting (see Section 2.3). While some stabilization criterion does not exceed a given threshold $\epsilon$, the two steps are repeated.

## 4   Soft-margin $\beta$-SVM

A major advantage of the empirical surrogate $\beta$-risk is that it can be plugged in numerous learning settings without radically modifying the original formulations. As an example, in this section we derive a new version of the Support Vector Machine problem, using the empirical surrogate $\beta$-risk , that takes into account the knowledge provided for each training instance (through the matrix $\beta$).

The soft-margin $\beta$-SVM optimization problem is a direct generalization of a standard soft-margin SVM and is defined as follows:

$$\arg\min_{\theta} \frac{1}{2}\|\theta\|_2^2 + c \sum_{i=1}^{m} \left(\beta_i^{\text{-}1}\xi_i^{\text{-}1} + \beta_i^{+1}\xi_i^{+1}\right)$$

$$s.t. \ \sigma(\theta^T \mu(x_i) + b) \geq 1 - \xi_i^\sigma \ \ \forall i = 1..m, \sigma \in \{-1, 1\}$$
$$\xi_i^\sigma \geq 0 \ \forall i = 1..m, \sigma \in \{-1, 1\}$$

where $\theta \in X'$ is the vector defining the margin hyperplane and $b$ its offset, $\mu : X \rightarrow X'$ a mapping function and $c \in \mathbb{R}$ a tuned hyper-parameter. In the rest of the paper, we will refer to $K : X \times X \rightarrow \mathbb{R}$ as the kernel function corresponding to $\mu$, i.e. $K(x_i, x_j) = \mu(x_i)\mu(x_j)$.

The corresponding Lagrangian dual problem is given by (the complete derivation is provided in the supplementary material):

$$\max_{\alpha} \ -\frac{1}{2} \sum_{i=1}^{m} \sum_{\substack{\sigma \in \\ \{\text{-1,+1}\}}} \sum_{j=1}^{m} \sum_{\substack{\sigma' \in \\ \{\text{-1,+1}\}}} \alpha_i^\sigma \sigma \alpha_j^\sigma \sigma' K(x_i, x_j) + \sum_{i=1}^{m} \sum_{\substack{\sigma \in \\ \{\text{-1,+1}\}}} \alpha_i^\sigma$$

$$s.t. \ 0 \leq \alpha_i^\sigma \leq c\beta_i^\sigma \ \ \forall i = 1..m, \ \sigma \in \{-1, 1\}$$
$$\sum_{i=1}^{m} \sum_{\substack{\sigma \in \\ \{\text{-1,+1}\}}} \alpha_i^\sigma \sigma = 0 \ \ \forall i = 1..m, \ \sigma \in \{-1, 1\}$$

which is concave w.r.t. $\alpha$ as for the standard SVM.

The $\beta$-SVM formulation differs from the SVM one in two points: first, the number of Lagrangian multipliers is doubled, because we consider both positive and negative labels for each instance; second, the upper-bounds for $\alpha$ are not the same for all instances but depend on the given matrix $\beta$. Like the coefficient $c$ in the classical formulation of SVM, those upper-bounds play the role of trade-off between under-fitting and over-fitting: the smaller they are, the more robust to outliers the learner is but the less it adapts to the data. It is then logical that the upper-bound for an instance $i$ depends on $\beta_i^\sigma$ because it reflects the reliability on the label $\sigma$ for that instance: if the label $\sigma$ is unlikely, its corresponding $\alpha_i^\sigma$ will be constrained to be null (and its adversary will have more chance to be selected as a support vector, as $\beta_i^\sigma + \beta_i^{-\sigma} = 1$). Also, those points for which no label is more probable than the other ($\beta_i^\sigma \rightarrow 0.5$) will have less importance in the learning process compared to those for which a label is almost certain. In order to fully exploit the advantages of our formulation, $c$ has to be finite and bigger than 0. As a matter of fact, when $c \rightarrow \infty$ or $c \rightarrow 0$, the constraints become exactly those of the original formulation.

## 5   Experimental Results

In the first part of this section, we present some experimental results obtained by adapting the iterative algorithm presented in Section 3 for semi-supervised learning and combining it with the previously derived $\beta$-SVM . Note that some approaches based on SVMs have been already presented in the literature to address the problem of semi-supervised learning. Among them, TransductiveSVM [5]

iteratively learns a separator with the labeled instances, classifies a subset of the unlabeled instances and adds it to the training set. On the other hand, WellSVM [14] combines the classical SVM with a label generation strategy that allows one to learn the optimal separator, even when the training sample is not completely labeled, by convexly relaxing the original Mixed-Integer Programming problem. In [14], WellSVM has been shown to be very effective and better than TransductiveSVM and the state of the art. For this reason, we compare in this section $\beta$-SVM to WellSVM. In the second subsection, we present some preliminary results in the noise-tolerant learning setting, showing how $\beta$-SVM behaves when facing data with label noise.

## 5.1 Iterative $\beta$-SVM for semi-supervised learning

We compare our method's performances to those of WellSVM, that has been proved, in [14], to performs in average better than the state of the art semi-supervised learning methods based on SVM and the standard SVM as well. In a semi-supervised context, a set $\mathcal{X}_l$ of labeled instances of size $m_l$ and a set $\mathcal{X}_u$ of unlabeled instances of size $m_u$ are provided. The matrix $\beta$ is initialized as follows:

$$\forall i = 1..m_l \text{ and } \forall \sigma \text{ in } \{-1, 1\}, \ ^0\beta_i^\sigma = 1 \text{ if } \sigma = y_i, 0 \text{ otherwise},$$

$$\forall i = m_l+1..m_u \text{ and } \forall \sigma \text{ in } \{-1, 1\}, \ ^0\beta_i^\sigma = 0.5$$

and we learn an optimal separator:

$$h^{t+1} = P_1(\mathcal{X}_l \cup \mathcal{X}_u, {}^t\beta) = \arg\min_h \ c_1 \mathcal{R}_\phi^{{}^t\beta}(\mathcal{X}_l, h) + c_2 \mathcal{R}_\phi^{{}^t\beta}(\mathcal{X}_u, h) + \mathcal{N}(h).$$

Here $c_1$ and $c_2$ are balance constants between the labeled and unlabeled set: when the number of unlabeled instances become greater than the number of labeled instances, we need to reduce the importance of the unlabeled set in the learning procedure because there exists the risk that the labeled set will be ignored. We consider the provided labels to be correct, so we keep the corresponding $_l\beta$ fixed during the iterations of the algorithm and estimate $_u\beta$ by optimizing $P_2(\mathcal{X}_u, h^{t+1})$. The iterative algorithm with $\beta$-SVM is implemented in Python using Cvxopt (for optimizing $\beta$-SVM ) and Cvxpy [2] with its Ecos solver [9].

For each dataset, we show in Figure 1 the accuracy of the two methods with an increasing proportion of labeled data. The different approaches are compared on the same kernel, either the linear or the gaussian, the one that gives higher overall accuracy. As a matter of fact, the choice of the kernel depends on the geometry of the data, not on the learning method.

For each proportion of labeled data, we perform a 4-fold cross-validation and we show the average accuracy over 10 iterations. Concerning the hyper-parameters of the different methods, we fix $c_2$ of $\beta$-SVM to $c_1 \frac{m_l}{m}$, $c_1$ of WellSVM to 1 as explained in [14] and all the other hyper-parameters ($c_1$ for $\beta$-SVM and $c_2$ for WellSVM) are tuned by cross-validation through grid search. As for the stopping criteria, we fix $\epsilon$ of $\beta$-SVM to $10^{-5} + 10^{-3}\|h\|_{\mathcal{F}}$ and $\epsilon$ of WellSVM to $10^{-3}$ and the maximal number of iterations to 20 for both methods. When using the gaussian kernel, the $\gamma$ in $K(x_i, x_j) = \exp(-\|x_i - x_j\|_2^2/\gamma)$ is fixed to the mean distance between instances.

Our method performs better than WellSVM, with few exceptions, and is more efficient in terms of CPU time: for the Australian dataset, the biggest dataset in number of features and instances, WellSVM is in average 30 times slower than our algorithm (without particular optimization efforts).

## 5.2 Preliminary results under label-noise

We quickly tackle another setting of the weakly labeled data field: the noise-tolerant learning, the task of learning from data that have noisy or uncertain labels. It has been shown in [3] that SVM learning is extremely sensitive to outliers, especially the ones lying next to the boundary. We study, the sensitivity of $\beta$-SVM to label noise artificially introduced on the Ionosphere dataset. We consider two initialization strategies for $\beta$: the *standard* on where $\beta^{y_i} = 1$ and $\beta^{-y_i} = 0$ and the *4-nn* one where $\beta^\sigma$ is set to the proportion of neighboring instances with label $\sigma$. In Figure 2, we draw the mean accuracy over 4 repetitions w.r.t. an increasing percentage (as a proportion of the smallest dataset) of two kinds of noise: the symmetric noise, introduced by swapping the labels of instances belonging to different classes, and the asymmetric noise, introduced by gradually changing the labels of the

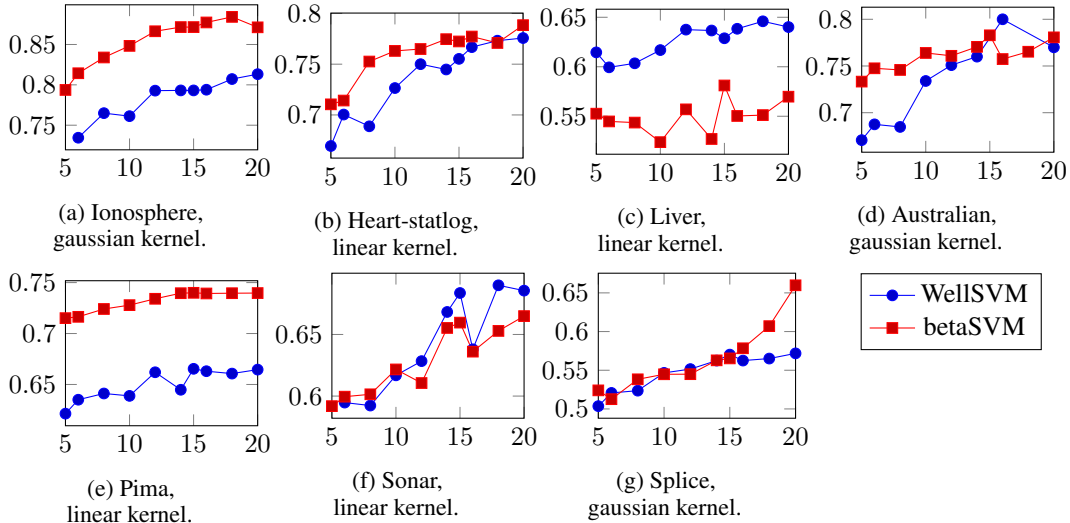

(a) Ionosphere, gaussian kernel.

(b) Heart-statlog, linear kernel.

(c) Liver, linear kernel.

(d) Australian, gaussian kernel.

(e) Pima, linear kernel.

(f) Sonar, linear kernel.

(g) Splice, gaussian kernel.

Figure 1: Comparison of the mean accuracies of WellSVM and $\beta$-SVM versus the percentage of labeled data on different UCI datasets.

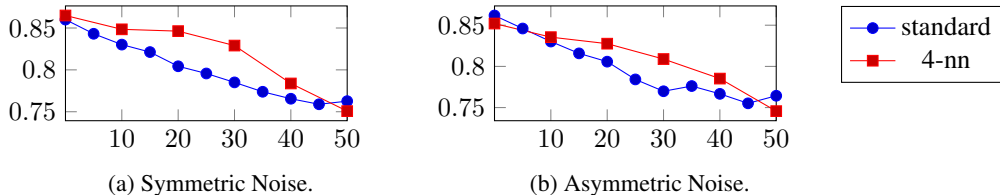

(a) Symmetric Noise.

(b) Asymmetric Noise.

Figure 2: Comparison of the mean accuracy versus the percentage of noise of iterative $\beta$-SVM with different initializations of $\beta$. The *standard* curve refers to the initialization of $\beta^{y_i} = 1$ and $\beta^{-y_i} = 0$ and the *4-nn* to the initialization of $\beta^{\sigma}$ to the proportion of neighboring instances with label $\sigma$.

instances of one class. These preliminary results are encouraging and show that locally estimating the conditional class density to initialize the $\beta$ matrix improves the robustness of our method to label noise.

# 6   Conclusion

This paper focuses on the problem of learning from weakly labeled data. We introduced the $\beta$-risk which generalizes the standard empirical risk while allowing the integration of weak supervision. From the expression of the $\beta$-risk , we derived a generic algorithm for weakly labeled data and specialized it in an SVM-like context. The resulting $\beta$-SVM algorithm has been applied in two different weakly labeled settings, namely semi-supervised learning and learning with label noise, showing the advantages of the approach.

The perspectives of this work are numerous and of two main kinds: covering new weakly labeled settings and studying theoretical guarantees. As proposed in Section 2.3, the $\beta$-risk can be used in various weakly labeled scenarios. This requires to use different strategies for the initialization and the refinement of $\beta$, and also to propose proper priors for these parameters. Generalizing the proposed $\beta$-risk to a multi-class setting is a natural extension as $\beta$ is already a matrix of class probabilities. Another broad direction involves deriving robustness and convergence bounds for the algorithms built on the $\beta$-risk .

# 7   Acknowledgments

We thank the reviewers for their valuable remarks. We also thank the ANR projects SOLSTICE (ANR-13-BS02-01) and LIVES (ANR-15-CE230026-03).

## Footnotes

[2]http://cvxopt.org/ and http://www.cvxpy.org/

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
