[Supplementary Material]

# $\beta$-risk: a New Surrogate Risk for Learning from Weakly Labeled Data
### Supplementary Material

**Valentina Zantedeschi**          **Rémi Emonet**

**Marc Sebban**

`firstname.lastname@univ-st-etienne.fr`
Univ Lyon, UJM-Saint-Etienne, CNRS, Institut d Optique Graduate School,
Laboratoire Hubert Curien UMR 5516, F-42023, SAINT-ETIENNE, France

## 1   Overview

This supplementary material is organized as follows: in Section 2, we prove the property of a Legendre conjugate of a permissible function used in Eq.(2) of Sec.(2) of the paper; in Section 3, we derive the dual problem of a soft-margin $\beta$-SVM;

## 2   Legendre Conjugate of Permissible Functions

The Legendre conjugate of a differentiable and strictly convex function $\phi$ can be written as:

$$\phi^*(x) = x\nabla_\phi^{-1}(x) - \phi(\nabla_\phi^{-1}(x)).$$

In the case of a permissible function $\phi$, its Legendre conjugate has the following property: $\phi^*(-x) = \phi^*(x) - x$.

*Proof.*

$$
\begin{aligned}
\phi^*(x) &= -x\nabla_\phi^{-1}(-x) - \phi(\nabla_\phi^{-1}(-x)) \\
&= -x(1 - \nabla_\phi^{-1}(x)) - \phi(1 - \nabla_\phi^{-1}(x)) \quad\quad (1)\\
&= -x + x\nabla_\phi^{-1}(x) - \phi(\nabla_\phi^{-1}(x)) \quad\quad\quad\quad (2)\\
&= \phi^*(x) - x
\end{aligned}
$$

Because of the symmetry of $\phi$ about $-\frac{1}{2}$, in Eq. (1) $\nabla_\phi^{-1}(-x) = 1 - \nabla_\phi^{-1}(x)$ and in Eq. (2) $\phi(1-x) = \phi(x)$. $\qquad\square$

## 3   Derivation of Soft-margin $\beta$-SVM

The soft-margin $\beta$-SVM optimization problem is a direct generalization of a standard soft-margin SVM and is defined as follows:

$$\arg\min_\theta \frac{1}{2}\|\theta\|_2^2 + c\sum_{i=1}^m \left( \beta_i^{\text{-}1}\xi_i^{\text{-}1} + \beta_i^{+1}\xi_i^{+1} \right)$$

$$s.t. \; \sigma(\theta^T\mu(x_i) + b) \geq 1 - \xi_i^\sigma \;\; \forall i = 1..m, \sigma \in \{-1, 1\}$$

$$\xi_i^\sigma \geq 0 \ \forall i = 1..m, \sigma \in \{-1, 1\}$$

where $\theta \in X'$ is the vector defining the margin hyperplane and $b$ its offset, $\mu : X \to X'$ a mapping function and $c \in \mathbb{R}$ a tuned hyper-parameter. In the rest of the paper, we will refer to $K : X^2 \to \mathbb{R}$ as the kernel function corresponding to $\mu$ ($K(x_i, x_j) = \mu(x_i)\mu(x_j)$).

Instead of solving the previous primal problem, it is preferable to solve its Lagrangian dual problem given by maximizing the corresponding Lagrangian w.r.t. its Lagrangian multipliers, which gives a nice Quadratic Programming problem that can be solved by common optimization techniques. The Lagrangian can be written as follows:

$$\mathcal{L}(\theta, b, \xi, \alpha, r) = \frac{1}{2} \|\theta\|_2^2 + c \sum_{i=1}^{m} \left( \beta_i^{-1} \xi_i^{-1} + \beta_i^{+1} \xi_i^{+1} \right) - \sum_{i=1}^{m} \sum_{\substack{\sigma \in \\ \{-1,+1\}}} \alpha_i^\sigma \left( \sigma(\theta^T \mu(x_i) + b) + \xi_i^\sigma - 1 \right) - \sum_{i=1}^{m} \sum_{\substack{\sigma \in \\ \{-1,+1\}}} r_i^\sigma \xi_i^\sigma$$

where $\alpha \in \mathbb{R}^{2*m}$ and $r \in \mathbb{R}^{2*m}$ are the Lagrangian multipliers. It is obvious that:

$$\max_{\alpha, r \geq 0} \min_{\theta, b, \xi} \mathcal{L}(\theta, b, \xi, \alpha, r) \leq \min_{\theta, b, \xi} \max_{\alpha, r \geq 0} \mathcal{L}(\theta, b, \xi, \alpha, r)$$

where the left term corresponds to the optimal value of the dual problem and the right one to the primal's one. The dual and the primal problems have the same value at optimality if the Karush-Kuhn-Tucker (KKT) conditions are not violated (see [1]). By setting the gradient of $\mathcal{L}$ w.r.t. $\theta, b$ and $\xi$ to 0, we find the saddle point corresponding to the function minimum:

$$\nabla_\theta \mathcal{L}(\theta, b, \xi, \alpha, r) = \theta - \sum_{i=1}^{m} \sum_{\substack{\sigma \in \\ \{-1,+1\}}} \alpha_i^\sigma \sigma \mu(x_i)$$

$$\nabla_b \mathcal{L}(\theta, b, \xi, \alpha, r) = -\sum_{i=1}^{m} \sum_{\substack{\sigma \in \\ \{-1,+1\}}} \alpha_i^\sigma \sigma$$

$$\nabla_{\xi_i^\sigma} \mathcal{L}(\theta, b, \xi, \alpha, r) = c\beta_i^\sigma - \alpha_i^\sigma - r_i^\sigma$$

which give

$$\theta = \sum_{i=1}^{m} \sum_{\substack{\sigma \in \\ \{-1,+1\}}} \alpha_i^\sigma \sigma \mu(x_i) \tag{3}$$

$$\sum_{i=1}^{m} \sum_{\substack{\sigma \in \\ \{-1,+1\}}} \alpha_i^\sigma \sigma = 0 \tag{4}$$

$$\alpha_i^\sigma \leq c\beta_i^\sigma \tag{5}$$

We can now write the QP dual problem by replacing $\theta$ by its expression (3) and simplifying following (4) and (5):

$$\max_\alpha \ -\frac{1}{2} \sum_{i=1}^{m} \sum_{\substack{\sigma \in \\ \{-1,+1\}}} \alpha_i^\sigma \sigma \sum_{j=1}^{m} \sum_{\substack{\sigma \in \\ \{-1,+1\}}} \alpha_j^\sigma \sigma K(x_i, x_j) + \sum_{i=1}^{m} \sum_{\substack{\sigma \in \\ \{-1,+1\}}} \alpha_i^\sigma$$

$$s.t. \ 0 \leq \alpha_i^\sigma \leq c\beta_i^\sigma \ \forall i = 1..m, \ \sigma \in \{-1, 1\}$$

$$\sum_{i=1}^{m} \sum_{\substack{\sigma \in \\ \{-1,+1\}}} \alpha_i^\sigma \sigma = 0 \ \forall i = 1..m, \ \sigma \in \{-1, 1\}$$

which is concave w.r.t. $\alpha$.

*Proof.*

$$\mathcal{L}(\alpha) = \frac{1}{2} \sum_{i=1}^{m} \sum_{\substack{\sigma \in \\ \{-1,+1\}}} \alpha_i^\sigma \sigma \mu(x_i) \sum_{j=1}^{m} \sum_{\substack{\sigma \in \\ \{-1,+1\}}} \alpha_j^\sigma \sigma \mu(x_j) + c \sum_{i=1}^{m} \left( \beta_i^{-1} \xi_i^{-1} + \beta_i^{+1} \xi_i^{+1} \right)$$

$$- \sum_{i=1}^{m} \sum_{\substack{\sigma \in \\ \{-1,+1\}}} \alpha_i^\sigma \left( \sigma \left( \left( \sum_{j=1}^{m} \sum_{\substack{\sigma \in \\ \{-1,+1\}}} \alpha_j^\sigma \sigma \mu(x_j) \right) \mu(x_i) + b \right) + \xi_i^\sigma - 1 \right) - \sum_{i=1}^{m} \sum_{\substack{\sigma \in \\ \{-1,+1\}}} r_i^\sigma \xi_i^\sigma \quad (6)$$

$$= -\frac{1}{2} \sum_{i=1}^{m} \sum_{\substack{\sigma \in \\ \{-1,+1\}}} \alpha_i^\sigma \sigma \mu(x_i) \sum_{j=1}^{m} \sum_{\substack{\sigma \in \\ \{-1,+1\}}} \alpha_j^\sigma \sigma \mu(x_j) + \sum_{i=1}^{m} \sum_{\substack{\sigma \in \\ \{-1,+1\}}} \alpha_i^\sigma$$

$$+ \sum_{i=1}^{m} \sum_{\substack{\sigma \in \\ \{-1,+1\}}} \left( c\beta_i^\sigma - \alpha_i^\sigma - r_i^\sigma \right) \xi_i^\sigma - b \sum_{i=1}^{m} \sum_{\substack{\sigma \in \\ \{-1,+1\}}} \alpha_i^\sigma \sigma \quad (7)$$

$$= -\frac{1}{2} \sum_{i=1}^{m} \sum_{\substack{\sigma \in \\ \{-1,+1\}}} \alpha_i^\sigma \sigma \sum_{j=1}^{m} \sum_{\substack{\sigma \in \\ \{-1,+1\}}} \alpha_j^\sigma \sigma K(x_i, x_j) + \sum_{i=1}^{m} \sum_{\substack{\sigma \in \\ \{-1,+1\}}} \alpha_i^\sigma \quad (8)$$

In Eq. (7) the third and the fourth terms are null because of (4) and (5). $\qquad \square$

We need the following two additional constraints in order to respect the KKT conditions which justify guarantee that the optimal value found by solving the dual problem corresponds to the optimal value of the primal:

$$\alpha_i^\sigma \left( \sigma(\theta^T + b) - 1 + \xi_i^\sigma \right) = 0 \, \forall \, i = 1..m, \sigma \in \{-1, 1\}$$

$$r_i^\sigma \xi_i^\sigma = 0 \, \forall \, i = 1..m, \sigma \in \{-1, 1\}$$

Once the Lagrangian dual problem solved, the characteristic vector $\theta$ and offset $b$ of the optimal margin hyperplane can be retrieved by means of the support vectors machine, i.e. the instances whose corresponding $\alpha_i^\sigma$ are strictly greater than 0:

$$\theta = \sum_{i)1}^{m} \sum_{\substack{\sigma \in \\ \{-1,+1\}}} \alpha_i^\sigma \sigma \mu(x_i)$$

$$b = \theta \mu(x_k) - \sigma_k$$

and the new instances can be classified :

$$y(x) = sign(\sum_{i=1}^{m} \sum_{\substack{\sigma \in \\ \{-1,+1\}}} (\alpha_i^\sigma \sigma K(x_i, x)) + b)$$

# 4 Additional Experiments

## 4.1 Semi-Supervised Learning

We report a table of mean accuracies with their relative errors of the performances of standard SVM, WellSVM and our method on 7 UCI datasets with 5%,10% and 15% of labeled instances of the training sets.

| dataset | % labeled | SVM | WellSVM | betaSVM |
|---|---|---|---|---|
| ionosphere | 5 | $0.74 \pm 0.02$ | $0.72\pm0.04$ | **0.77±0.03** |
|  | 10 | $0.78 \pm 0.03$ | $0.79\pm0.03$ | **0.80±0.04** |
|  | 15 | $0.81 \pm 0.01$ | **0.82±0.02** | $0.81\pm0.02$ |
| sonar | 5 | $0.58 \pm 0.06$ | $0.58\pm0.03$ | **0.59±0.05** |
|  | 10 | $0.65 \pm 0.04$ | $0.64\pm0.04$ | **0.66±0.05** |
|  | 15 | $0.65 \pm 0.02$ | **0.67±0.02** | **0.67±0.02** |
| liver | 5 | **0.59±0.02** | $0.51\pm0.04$ | $0.55\pm0.04$ |
|  | 10 | **0.61±0.04** | $0.54\pm0.03$ | $0.58\pm0.03$ |
|  | 15 | **0.64±0.04** | $0.54\pm0.03$ | $0.58\pm0.03$ |
| splice | 5 | **0.53±0.07** | $0.50\pm0.07$ | **0.53±0.06** |
|  | 10 | **0.56±0.02** | $0.55\pm0.05$ | $0.55\pm0.07$ |
|  | 15 | **0.60±0.03** | $0.56\pm0.05$ | $0.56\pm0.04$ |
| heart-statlog | 5 | $0.64\pm0.04$ | $0.55\pm0.03$ | **0.71±0.04** |
|  | 10 | $0.72\pm0.03$ | $0.62\pm0.02$ | **0.76±0.03** |
|  | 15 | $0.73\pm0.02$ | $0.63\pm0.03$ | **0.77±0.02** |
| australian | 5 | $0.72 \pm 0.05$ | $0.64 \pm 0.01$ | **0.73±0.06** |
|  | 10 | **0.73±0.03** | $0.72 \pm 0.04$ | **0.73±0.04** |
|  | 15 | **0.76±0.07** | $0.75 \pm 0.03$ | $0.75 \pm 0.04$ |
| pima | 5 | $0.65\pm0.01$ | $0.62\pm0.03$ | **0.71±0.01** |
|  | 10 | $0.69\pm0.01$ | $0.63\pm0.03$ | **0.72±0.01** |
|  | 15 | $0.71\pm0.01$ | $0.64\pm0.03$ | **0.72±0.01** |

## 4.2 Robustness to Label Noise

Here we report the results of applying $\beta$-SVM to a synthetic dataset and study its robustness to artificially induced label noise.

The synthetic dataset consists in 40 instances of 2 balanced classes: the instances of each class are uniformly distributed around a center point so that they can be easily classified by a linear separator to which we will refer as the true separator.

In Fig. 1, we compare the linear classifiers learned at each iteration of our iterative, algorithm with $\beta$-SVM, with a standard linear SVM and with the true separator. We conducted the experiment as follows: we apply the two methods first on the original dataset, then on a dataset where we swapped the label of a random instance of each class and so on with an increasing number of swapped labels.

We notice that our method is more robust to label noise: even though at the first iteration, we learn the same separator as the standard linear SVM, through the following iterations the algorithm converges to a separator closer to the true separator.

## References

[1] S. Boyd and L. Vandenberghe. *Convex optimization*. Cambridge university press, 2004.

Figure 1: Artificially induced label noise: the baseline, here, corresponds to the separator learned with a classical SVM. The first figure shows the learned separators with the original labels, and the other figures show the results for an increasing number of swapped labels going from left to right and from to bottom.