[Reviews · NeurIPS 2016]

Reviewer 1

Summary

The paper proposes a surrogate risk for weakly supervised learning. The risk allows for a per example weighting which can be adapted to different learning scenarios. A condition is determined under which the proposed risk is a tight bound to the 01 risk. This is used to derive an iterative optimisation algorithm. Experiments are presented for semi-supervised and noisy label learning.

Qualitative Assessment

UPDATE AFTER REBUTTAL: I still feel the value of the framework isn't fully convincing. My basic issue is that for the weakly supervised scenarios that don't already have principled algorithms, the precise reason the provided formulation is superior is unclear. For example in SSL the proposed method has a very similar flavour to self-training. I like however that there is an attempt at an unified approach to a range of such problems, and it's possible that one could do something interesting with this framework in future work. On the technical side, I still don't quite get the optimisation proposed for beta (Line 144). As I understand, Eqn 4 implies R_\phi^\beta(X, h) = R_\phi(S, h) + (1/m) sum_{i = 1}^m beta_i y_i h(x_i) so it seems like optimising R_\phi^\beta(X, h) is equivalently optimising the weighted mean term. But as that's constrained to be zero in the optimisation, we're just finding any feasible beta? The authors comment that there is a constraint on beta that is needed for the above equality, but this constraint seems indeed imposed in the optimisation. The basic idea of constructing new risks that can adapt to different learning scenarios is interesting, and in keeping with some recent work for special cases e.g. Patrini et al. 2013 for learning from label proportions. The paper begins with a decomposition of an arbitrary supervised risk into a weighted unsupervised risk, plus an additional term which is a certain beta-weighted mean. The idea is then that if one can somehow omit the second term, one can learn using the first term alone, which is purely unsupervised. The precise decomposition given in Lemma 1 is certainly interesting, and appears novel, but seems a simple extension of Thm 3 of Patrini et al. 2013 (specifically, the result is implicit in the proof of said theorem), or Lemma 1 of Patrini et al. Loss factorization, weakly supervised learning and label noise robustness. ICML 2016. Thus, I think the key issue is whether the result is compellingly used for new weakly supervised learning problems. On this front, I think the paper perhaps falls a bit short. It starts with a nice observation, which perhaps is intended to be the key insight of the paper: to make the new beta risk and the original as close as possible, one must constraint the beta-weighted mean of the classifier to be zero. This appears a departure from previous work, and gives a concrete criterion to learn from weakly supervised data. However, this term depends on the unobserved labels for the instances, and so the proposed method unavoidably runs into the issue that purely learning from unlabelled data is impossible without further assumptions. The chosen approach to get around this is, unsurprisingly, a heuristic: one guesses the labels for the instances based on the current model (reminiscent somewhat of self-training for semi-supervised learning). The framework would have been more compelling were there something cleverer that could be done here. Given that a heuristic is employed, I think the paper could have explained in more detail why the proposed scheme is more attractive than existing proposals (unified framework?). With this, the paper seeks to iteratively optimise for a good classifier, and the set of per instance weights beta. The optimisation of the classifier is standard; for the beta weights, one seeks to minimise the new risk subject to the constraint that the weighted mean is zero. while the latter is intuitive, I was unsure as to the well posedness of this optimisation. In the second stage of the iterative algorithm of section 3, the object that is optimised is R(h, beta; X). As I understand, section 2.2 establishes that this is equivalent to R(h, beta; S) + mu(beta), where mu(beta) is the weighted mean that is constrained to 0. But isn't R(h, beta; S) further just R(h; S) by Lemma 2.1, in which case the objective is effectively mu(beta), as the other term is independent of the precise choice of beta? In which case we would simply pick any admissible beta which satisfied the constraint? (This is potentially ok, as it doesn't affect the final risk? But the issue seems unclear.) The experiments presented are promising, showing the potential to improve upon the WellSVM in certain cases. Overall, the paper has an interesting perspective on weakly supervised learning, albeit one that builds upon similar proposals for special cases in Patrini et al. The proposed method is simple, and seems to perform well in practice. However, the method perhaps falls short of being a theoretically justified, unified solution to weakly supervised learning: it unavoidably faces the problem of having to make assumptions to get around the lack of labels, and does so in an acceptable if not particularly insightful way. Other comments: - it might be more standard to use \ell instead of \mathcal{L} for the loss function - consider using e^-x or exp(-x) instead of exp^{-x} - explicitly show dot product or transpose in definition of Bregman divergence - the discussion about seeking an upper bound to the 01 risk could perhaps comment on the further need for a regret bound between the 01 and the newly proposed risk. Otherwise, the optimisation would have no guarantee of actually converging to the correct solution. - the notation ^t\beta for the iterates seems awkward to me. - It is stated that one estimates labels based on the current set of weights beta. in the case of semi-supervised learning, the weights are initialised to 0.5. How does one pick weights in this case? Is it a random flip? - kernel should be on X x X, not X^2? - the proposed method requires that one learns a separate wait for every instance. When the number of instances is large, this might be problematic. Can you comment on possible ways of alleviating the burden? - consider having markers for the points displayed in the plots. - for citation [13], capitalize SVMs.

Confidence in this Review

2-Confident (read it all; understood it all reasonably well)


Reviewer 2

Summary

The paper performs weighting of the loss with the class posterior probabilities, in semi-supervised and noise-tolerant learning.

Qualitative Assessment

Pros: - A nice, in some sense novel, way of introducing responsibilities into the surrogate loss. - An argument that this loss is a reasonable upper bound on the 0-1 loss Cons: The restriction of wellSVM to SVMs, and not other surrogate losses, and the fact that one has to derive new optimization approaches for every new task seems minor issues. No theoretical or conceptual issues from an actual learning/generalization perspective are raised. So, even though maybe not straightforward to apply, wellSVM in principle solves what needs to be solved? While the idea of the paper is interesting (presented in sections 1-2.2), the authors do not offer much in the way of explanation as to why this approach works and how it relates to other, similar approaches to semi-supervised learning. The most interesting part of the new objective function is not the weighted version of the losses (R^\beta_\phi), which has been studied before in the literature, but the constraint, which seems to be the novel part of the formulation. I think it would improve the manuscript if the authors would explain why this constraint is expected to make a large difference compared to other methods. Without the constraint, the approach just seems to be an iterative procedure similar to some training procedures for TSVM, EM, self-learning/pseudo-labeling, etc. In line 98, because the labels have to be estimated, do we not run the risk that the surrogate loss is no longer an upper bound because the true ‘sum term’ (rather than the estimated one) can become negative? Some statements in the paper are too strong or not properly explained, for instance "guarantees that minimizing them also minimizes the 0-1 loss. “ and "indisputably allowed us to increase the confidence in the models induced by machine learning methods.” which I would, in fact, dispute. Also, some things are made more complicated then they are, like calling \sigma "a Rademacher variable” while it does not have the equal probability attached to it that I would associate with a Rademacher variable. At a minimum, I think the experimental results can be improved by adding the performance of the supervised SVM for comparison. The noise-tolerant setting is a bit incomplete to me. Instead I would prefer the authors to extend their discussion of the semi-supervised part of the paper. Some detailed comments: - 2.1) " [other loss functions are] ... smooth relaxations of the 0-1 loss and are all defined in a way that guarantees that minimizing them also minimizes the 0-1 loss." This is not true. - 2.1) ".. a permissible function is a function f: [0,1] -> R^-, symmetric about -1/2..." Symmetric in its image? - 2.1) Where is the inverse gradient coming from? - 2.2) What is the domain of the betas? Non-negative, in [0,1], sum to 1? - 2.2) Hypothesis space is used but not mentioned. - 2.2) Sigma is used but not mentioned. - 2.2) "... (ii) to be sufficient to learn without accessing the labels." What do you mean by this? Unsupervised learning? - 2.2) Enforcing the constraint, sum_i^m beta_i^-y_i y_i h(x_i) = 0, in an imbalanced class problem would be problematic, right? As you would be assigning half of all your points to 1 class? - 2.3) I don't think MIL fits in your weakly labeled setting - 3) It is not mentioned what this N-regularization entails. L-p norm? - 4) The purpose or form of this mu-function is not explained. - 4) ".. mu(K(x_i,x_j)) = mu(x_i)mu(x_j).." Can you give an example of a kernel function that fulfills this constraint? - 4) ".. K : X^2 -> R" X^2 denotes two-dimensional data. I think you mean X \times X - 4) How is it concave in alpha? - 5) Typo: "convexly" - 5.1) "..X_u.." Inconsistent notation, {\cal X}_u is later on - figure) There is no difference between the plots in black-white - 5.1) Why are there two balance constants? What are the constraints on c1,c2?

Confidence in this Review

2-Confident (read it all; understood it all reasonably well)


Reviewer 3

Summary

The manuscript proposes a new beta-surrogate risk for learning in weakly-supervised scenarios such as label noise, multiple-instance learning, semi-supervised learning and the like. The formulation is novel and elegant, the paper is well-written. I have a few questions for the authors, but I like the paper overall and recommend acceptance.

Qualitative Assessment

The manuscript proposes a new beta-surrogate risk (the corresponding loss interpreted as a surrogate of the classical 0-1 loss) for learning in weakly-supervised scenarios including label noise, multiple-instance learning, semi-supervised learning and the like. The formulation captures existing solutions as special cases, and also instantiates to give new methods in some cases. As far I can see, the formulation is novel. The paper is well-written. There are a few questions for the authors and some suggestions listed below: (i). Two relevant papers that I see missing from the references: 1. Learning with noisy labels. N Natarajan, IS Dhillon, PK Ravikumar, A Tewari. NIPS 2013. 2. A convex relaxation for weakly supervised classifiers. A Joulin, F Bach. ICML 2012. In [1], the authors propose a label-weighted surrogate for 0-1 loss in the presence of label noise. I believe this would be a special case of the proposed \beta-risk formulation. It would be good to add a discussion of this. More importantly, the convex formulation in [2] also is in the similar spirit as this submission, in catering to different weakly-supervised scenarios, especially semi-supervised learning and multiple-instance learning. This is a key citation missing from the paper. I would like the authors to comment on how the two approaches differ. In particular, is the proposed \beta-risk formulation more general than [2]? (ii) It is a bit unsatisfying that the authors do not give insights into the proposed heuristic iterative algorithm that alternates between updating beta and the model parameters. For example, it would be good to design synthetic experiments, and examine the converged beta values. The authors talk about sparse beta prior for multiple-instance learning, but I highly doubt if it would be effective in practice. This problem plagues many generalization frameworks. Even in [1] (learning with noisy labels), the problem of estimating weights does not have a satisfactory solution as yet. Unfortunately, the manuscript does not make any progress towards this. Could the authors comment on this? (I've read the rebuttal.)

Confidence in this Review

2-Confident (read it all; understood it all reasonably well)


Reviewer 4

Summary

This paper proposes a new surrogate risk for learning from weakly labeled data. The main idea is to introduce a hyper-parameter beta to define the loss function. This beta could be regarded as capturing a nature behind data points in many weakly labeled data learning. The paper proposes iterative algorithm for beta-risk: it first solves risk minimization problem under fixed beta, then optimizes the hyper-parameter beta to satisfy the condition for the beta.

Qualitative Assessment

The idea is interesting. One concern is the choice of the hyper-parameter beta sounds highly problem dependent, so the risk require the well-designed initialization for every different weakly labeled learning, e.g., multi-instance learning and noise-tolerant learning. If that concern is clarified, the paper would make the readers are confident to the method. It seems that there are some mistakes in equations. I think however it fortunately would not affect the algorithm in Section 3. In page 3, the line of the definition of the risk R_phi(S, h), since R_phi is the surrogate loss, from Lemma 1 in [15], the equality from first term and second term does not hold (if \nabla^{-1} is removed, it will be hold). For first term and third term, if one remove b_phi on third term, it will be hold (see Lemma 3 in [15]). Thus subsequent derivations should be changed. Although the above mistakes exist, the algorithm would not be affected of it because b_phi is finally replaced with the hyper-parameter c. The minor comments are the followings: 1. In page 2, the definition of the permissible function in the last line, the range of f is R^+, and it is symmetric about +1/2. 2. The number of m_l and m_u used in the experiments should be reported. It will help other researches to confirm your results.

Confidence in this Review

2-Confident (read it all; understood it all reasonably well)


Reviewer 5

Summary

Well-described by the abstract. The authors introduce a new surrogate risk. This surrogate is a generalization of classical empirical risk, with a tunable parameter beta that, when >0 allows for the injection of side-information about labels (& thus handling weak labels).

Qualitative Assessment

The exposition & theoretical results seem reasonable, and to my knowledge are novel. These are, however, aimed at achieving superior semi-supervised learning results. This is where most of my criticism would be. Empirical results don't seem strong. Based on the data available (plots), beta-SVM is comparable in most cases, clearly worse in one and clearly better in around 2. Given dataset sizes (~300 examples for most), and showing results for labeling less than 20%, statistical significance of most differences is questionable. For example, a 0.1 difference in accuracy over 30 examples has a ~0.1 p-value.

Confidence in this Review

1-Less confident (might not have understood significant parts)


Reviewer 6

Summary

This paper presents a new method of semi-supervised learning. The loss function is split into two terms, one that includes the labels and one that does not. The two terms are balanced by "beta" variables that specify the level of confidence per-sample. This formulation generalizes normal risk minimization, which is a special case when one of the betas for a sample is set to 1 (100% confidence in that label). This is used for labeled samples and the betas are not updated. For unlabeled samples, the betas are iteratively updated in such a way that the beta-risk is minimized, while still guaranteeing that the beta-risk provides an upper-bound of the 0-1 loss. The method is general to all well-defined surrogates, although provides derivations and experiments only for the soft-margin SVM. The paper compares this to a leading method, WellSVM (Li et al.), on seven UCI datasets with labels known for only only 5-20% of the data. It also shows application on a separate task, where all the data is labeled but with some level of labeling error.

Qualitative Assessment

This is an interesting and promising method, but what really needs to be strengthened is the experimental comparison with WellSVM. The WellSVM paper reports results on 16 UCI datasets, with error bars and a quantitative summary. This paper only covers 7 datasets and there are no error bars nor quantitative summary. From the WellSVM paper, it is clear that the error bars on these problems are generally big compared to the gap between WellSVM and betaSVM. A table similar to Tab. 2-4 in the WellSVM paper, where betaSVM is added and shown to outperform WellSVM would make this paper much stronger. This is the only reason for the lower impact score, but I think this could easily be improved by including more datasets and thus showing a statistically clear improvement over WellSVM. There are also discrepancies between the values that you report and the values that WellSVM report and I am not sure why. Both papers seem to have similar experimental setups and do comparable hyper-parameter searches, including considering both linear and gaussian kernels. For instance, WellSVM on "Australian" at 5% labeled data is reported by the WellSVM paper to get 81±4% accuracy, while this paper reports it as ~67%.

Confidence in this Review

1-Less confident (might not have understood significant parts)